# Controversial Impact of Vitamin D Supplementation on Reducing Insulin Resistance and Prevention of Type 2 Diabetes in Patients with Prediabetes: A Systematic Review

**DOI:** 10.3390/nu15040983

**Published:** 2023-02-16

**Authors:** Agata Pieńkowska, Justyna Janicka, Michał Duda, Karena Dzwonnik, Kamila Lip, Aleksandra Mędza, Agnieszka Szlagatys-Sidorkiewicz, Michał Brzeziński

**Affiliations:** Department of Pediatrics, Gastroenterology, Allergology and Pediatric Nutrition, Medical University of Gdansk, 80-803 Gdansk, Poland

**Keywords:** prediabetes, vitamin D, insulin resistance, type 2 diabetes

## Abstract

Background: Prediabetes has become a worldwide health problem. Multiple clinical trials have been conducted to determine the potential benefits of vitamin D supplementation in preventing the conversion to diabetes, but the results are inconsistent. The aims of this study were to evaluate the current knowledge and to suggest recommendations for researchers on designing future trials regarding that matter. Methods: Four databases were searched for randomized control trials from the last 10 years about vitamin D and insulin resistance. The systematic electronic literature search identified 2645 studies, of which thirty-eight qualified for full-text reading and discussion. Finally, eight trials were included. Results: Final results of seven trials reported that supplementation of vitamin D does not reduce insulin resistance nor reduces the risk of diabetes mellitus type 2 development in prediabetes. Only one trial showed improvements in fasting glucose and HOMA-IR. Conclusions: Due to the great variation and biases in study designs, an unambiguous interpretation of the results is not possible. To eliminate those vulnerabilities in the future, we made certain suggestions for study design. Long-term and well-designed studies are still required.

## 1. Introduction

A significant change regarding the nature of diseases causing disability or premature death has been observed. Chronic diseases such as diabetes, often linked with progressively changing lifestyles, have replaced contagious diseases and are especially visible in western societies [1,2]. Type 2 diabetes is prognosed to become an increasingly prevalent health problem⁠ [3]. Although lifestyle change is the primary method of prevention, inevitably, other solutions to improve patients outcome are considered.

Prediabetes has become one of the leading health problems worldwide [4] in recent decades. While no universally accepted definition exists, prediabetes is most commonly described as a state of intermediate hyperglycemia with glycemic parameters higher than normal but below the level indicative of diabetes. World Health Organization (WHO) lists those parameters as impaired fasting glucose (IFG), defined as fasting plasma glucose (FPG) ranging between 6.1 to 6.9 mmol/L (110–125 mg/dL); and impaired glucose tolerance (IGT), defined as plasma glucose level of 7.0–11.0 mmol/L (126–200 mg/dL) measured 2 h after ingestion of 75 g of oral glucose load (OGGT) [5]. The American Diabetes Association (ADA) criteria, however, while having the same limits for IGT, use different cut-off values for IFG (FPG 5.6–6.9 mmol/L (100–125 mg/dL) [6].

The term “prediabetes” is considered to be controversial itself for several reasons. First of all, it suggests that all prediabetic patients will eventually develop diabetes, which is not the case [7]. Secondly, the „pre” prefix suggests the absence of a disease, which may delay intervention. Lastly, there are risk factors other than intermediate hyperglycemia, which may result in diabetes [8,9,10].

Despite all the controversies regarding terminology, prediabetes has been proven to be a cause of various complications typically associated with long-term diabetes [11,12]. Because of this, many possible interventions for reversing prediabetes have been suggested [13], one of which is vitamin D supplementation.

Vitamin D is a fat-soluble vitamin, crucial for maintaining constant extracellular calcium ion levels and the maintenance of calcium and phosphorus homeostasis [14]. It is the only vitamin that can be acquired not only through nutrition but also synthesized in the skin during exposure to UV radiation [15,16]. According to the data provided by scientific organizations, e.g., the World Medical Association [17] or the Institute of Medicine, Food and Nutrition Board [18], vitamin D deficiency is considered to be at a level below 50 nmol/L or 20 ng/mL. Severe vitamin D deficiency with a 25(OH)D concentration below <30 nmol/L (or 12 ng/mL) is considered to be health-threatening [19,20]. Hipovitaminosis is most commonly associated with rickets in children [21,22] or osteomalacia in adults [23], vitamin D importance in maintaining the homeostasis of other systems has also been suggested [24,25,26]. Correlations between low vitamin D status and diabetes and other conditions comprising metabolic syndrome have also been noted [25,27], and thus questions about vitamin D supplementation as a possible way of reversing prediabetes, or at the very least delaying its conversion into type 2 diabetes, have been raised [28,29].

Multiple clinical trials have been conducted to determine the potential benefits of vitamin D supplementation in prediabetic patients for delaying or preventing the conversion to diabetes, but the results so far have been inconclusive at best. In this systematic review, randomized control trials (RCTs) from the last 10 years were analyzed and suggestions for future study designing were made. The aims of this study were to evaluate the current knowledge gathered from years of research on whether vitamin D supplementation reduces the progression of type 2 diabetes and to suggest recommendations for researchers on designing future trials regarding that matter.

## 2. Materials and Methods

A systematic review of randomized controlled trials, published between January 2012–September 2022, was performed. It was based on the PRISMA (Preferred Reporting Items for Systematic Reviews and Meta-Analyses) recommendations [30] in order to improve and standardize this study. The development of the research procedure began with the elaboration on the clinical question, “Does vitamin D supplementation reduce insulin resistance in prediabetic patients compared to placebo?” in accordance with the PICO model. A comprehensive literature search was performed using four databases: PubMed, Web of science, EBSCO, and Ovid, using two different search strategies in order to obtain more comprehensive results. The first strategy utilized multiple search terms listed in the Appendix A. The second strategy involved using only the key pair of keywords for our study: “vitamin D” and “prediabetes”. Duplicates were removed after the results were combined. After comparing the results from both sources, we found that despite using similar search terms, there was little overlap between the manuscripts we found utilizing both strategies.

We searched for randomized controlled trials, meta-analyses, and systematic reviews, from the last 10 years, published in English up to 17 September 2022. The following inclusion criteria were applied:participants with prediabetes, over 18 years of age, without other restrictions of age, sex, or ethnicity;vitamin D supplementation in the chemical forms of calciferol-D2 or cholecalciferol-D3 in any dose, administered with any frequency and with any time of follow up;primary outcome defined as the development of type 2 diabetes or its effect on insulin resistance;secondary outcomes included measurements of any conventional marker of glycemic control (fasting glucose, oral glucose tolerance test, insulin secretion, HOMA-IR index, glycated hemoglobin level) [31]

Exclusion criteria included co-supplementation, pregnancy, and studies in which the research group combined people with prediabetes and diabetes. The literature review was conducted using Rayyan (a web application facilitating systematic review) [32]. We also used Rayyan to remove duplicates and resolve conflicts over the inclusion of the articles.

The detailed research scheme is shown in Figure 1. In total, 5136 scientific papers were found using both methods of searching. After removing duplicates, the final database consisted of 2645 articles, which were included for their title and abstract review. Due to time constraints, each researcher could not read all the articles’ abstracts. This database was thus divided into three parts, each of which was assigned to two different researchers. Reading the same article abstracts by two independent investigators was intended to eliminate bias. Conflicts over whether a given article met the inclusion criteria were resolved by consensus.

During the next step, 2573 studies were excluded and the databases from both search strategies were combined. Meta-analyses and systematic reviews were separated from individual randomized control trials and searched by bibliography. It enabled us to include eight more RCTs in the final manuscript list. Once again, duplicates were removed. The final database consisted of 38 studies, which qualified for full-text reading, general analysis, and discussion. The inclusion and exclusion criteria described above were reapplied, as not all abstracts contained detailed descriptions of the methodology. In addition, manuscripts with lifestyle modifications or variable doses of vitamin D used in one trial were also excluded. Following this stage, eight papers [33,34,35,36,37,38,39,40] were selected for the final systematic review.

## 3. Results

The summary characteristics of the included trials are shown in Table 1. The studies we selected were conducted in various climate zones and in three different continents: North America, Europe, and Asia. Two of the studies were multicenter studies. The sample sizes ranged from 66 to 2423, although in all studies, the final analyzed group was smaller due to compliance issues. All participants were diagnosed with prediabetes based on the same criteria, which were consistent with the ADA guidelines for diagnosing prediabetes and diabetes. Seven studies used two or three criteria, i.e., fasting serum glucose or impaired glucose tolerance, or glycated hemoglobin. Only 1 article considered a single criterion, i.e., fasting blood sugar, to diagnose prediabetes [35].

The main demographic characteristics reported in the studies included sex, age, and ethnicity. Six trials recruited wide-age groups ranging from 18–25 years old to 75–80 years old [33,34,35,36,39,40], one trial recruited participants older than 40 years of age [38], and one trial recruited participants over 60 years old [37]. All trials, included both male and female participants, and all participants were overweight or obese. In two trials, researchers did not take into account the baseline vitamin D levels [34,40], and the remaining six included people with hypovitaminosis or suboptimal levels of vitamin D. One article did not report any specific exclusion criteria [38]. The remaining seven reviewed literature listed exclusion criteria such as diabetes mellitus, other diseases, or the use of medications that could affect glucose metabolism (in some papers, authors detailed specific diseases). Other exclusion criteria differed between studies.

In all selected studies, the only intervention was vitamin D supplementation. Different supplementation doses and different schedules of supplementation were used, including daily from 1000 IU to 4000 IU and weekly from 20,000 IU to 88,000 IU. All eight trials applied unchanging doses of supplementation and dosing frequency. The exception was a study in which participants were divided into two separate groups with two different dosage regimens, however, they were still analyzed separately [37]. The duration of the follow-up in the trials ranged from three months to five years, but only two lasted longer than a year. 

Not all trials used the same primary outcome. Only two studies focused on comparing the percentage of participants developing type 2 diabetes while supplementing vitamin D [34,36]. The others merely compared glucose metabolism parameters. Fasting glucose was assessed in all studies and OGTT was an important outcome in 7 out of 8 trials. The third most frequently assessed outcome was insulin resistance level (HOMA-IR). Some trials measured glycated hemoglobin, insulin secretion or beta-cells function. Two of the studies measured all of these parameters [38,39]. 

The final results of seven out of the eight analyzed trials showed that vitamin D supplementation neither significantly improved glucose metabolism nor reduced insulin resistance nor reduced the risk of type 2 diabetes developing in prediabetic subjects. Only one trial showed improvements in fasting glucose and HOMA-IR [35]. However, it was also the study with the shortest follow-up.

Despite considerable differences in the characteristics of participants, the intervention, and the assessment of primary and secondary outcomes, most of the studies similarly concluded that vitamin D supplementation had no significant effect on insulin resistance and the prevention of diabetes in patients with prediabetes.

## 4. Discussion

The purpose of our review was to evaluate the current state of knowledge on the effects of vitamin D supplementation in individuals with prediabetes, as well as attempt to establish guidelines for future studies. Although initially, the hypothesis that vitamin D supplementation might prevent the development of T2D seemed very promising [41], after years of research, the results are still inconsistent. In compiling this systematic review, we have identified studies that demonstrated that vitamin D supplementation reduced insulin resistance, did not reduce insulin resistance, or that it reduced it only in those individuals with a significant pre-existing vitamin D deficiency [35,39,42]. There were significant differences in the methodology of reviewed randomized controlled trials, which made it difficult to perform this review and draw uniformly applicable conclusions for all studies.

Regardless of the diversity in the study designs, however, there were some shared elements. In all reviewed studies, the patients were overweight or obese to begin with, with no weight reduction observed during the trial. This confirms that vitamin D supplementation itself does not result in weight reduction [43], and additional lifestyle modifications are required to obtain weight loss. Importantly, as we consider the effects of vitamin D on adipose tissue percentage, no lifestyle modification plan was applied in any of the reviewed studies. 

Another common factor was that despite different dosage regimens and characteristics of patients, in almost all studies, vitamin D supplementation normalized serum 25(OH)D levels. In one trial [35], the patients were given 1000 IU, which turned out to be an insufficient dose for overweight participants, as only 37% of the participants achieved the optimum level of serum 25(OH)D. This suggests that it is important in future research to administer sufficient doses consistent with recommendations for a given population to avoid difficulties in interpreting and comparing the trial results. 

While reviewing the literature, significant differences in the number of participants were noticed. Randomized control trials with a small number of participants and wide differences in baseline concentrations of vitamin D may be underpowered to detect small to moderate effects on outcomes [43]. This issue could be solved by standardizing the serum 25(OH)D data and RCT’s methods. It would provide the summed results from individual studies to conduct extensive meta-analyses.

According to the available data, 5–10% of patients with prediabetes become diabetic annually [7], so a long follow-up is required in order to assess vitamin D supplementation effects on the development of type 2 diabetes mellitus. Most of the assessed studies had an observational time of under one year, up to 26 weeks. In three of the trials, the follow-up period exceeded 12 months, and two of those had the development of T2D as the primary endpoint. In these studies [34,36], vitamin D supplementation did not prevent the progression from prediabetes to diabetes. These trials assessed different populations, used different dosages of vitamin D, and vitamin D deficiency was not one of the inclusion criteria. The last of the three trials with a longer follow-up [38] examined vitamin D supplementation in vitamin-D-deficient patients, measuring glycemic markers such as FBG, 2-h glucose levels in OGTT, and HbA1c (ADA criteria for prediabetes). The vitamin D dosages and study populations in this study were different from the previous two studies. No significant changes in glycemic markers were observed following vitamin D supplementation. 

In conclusion, it is worth emphasizing that when designing trials testing vitamin D supplementation, researchers should select only participants with hypovitaminosis D to avoid confusion in results, and the follow-up should be a minimum 12 months’ long to assess the actual impact of supplementation on the development of diabetes mellitus.

All studies included in our systematic reviews had ethics committee approvals, while control groups had patients with vitamin D deficiency. However, ethical concerns related to long-term non-supplementation of vitamin D in patients for the purposes of a research study should be considered. As mentioned above, patients are not completely deprived of vitamin D as it comes from food and sun exposure. Moreover, participants with a vitamin D concentration of 30 nmol/L–50 nmol/L may be included, which is still insufficient but is not life-threatening.

Only one of the analyzed trials showed a positive effect of vitamin D on decreasing insulin resistance [35]. The participants were given 1000 IU of vitamin D daily for 3 months, and FBG, fasting insulin, HOMA-IR, body fat percentage, and serum vitamin D level were measured. After 3 months of supplementation, a decrease in insulin resistance (HOMA-IR) and no changes in body fat percentage were observed. The first limitation we noticed in the study design was an insufficient dosage of vitamin D, as only 37% of participants reached an optimal level of serum vitamin D, which might have affected the inconsistency of the results. Another problem was the short duration of the trial. As mentioned above, a long observation period is required to determine the influence of vitamin D on diabetes mellitus development, and short-term changes in glycemic parameters are not a reliable indicator of that. The primary outcome measured in this study was insulin resistance, a surrogate rather than a clinical outcome, and HOMA-IR specifically is not a strong predictor of developing diabetes mellitus in the future [44]. HOMA-IR is also associated with other conditions, such as cardiovascular disease, nonalcoholic fatty liver disease, metabolic syndrome, and polycystic ovary syndrome or obesity. Research is still being conducted to estimate reliable norms for HOMA-IR [45,46]. The gold standard for insulin resistance assessment is a hyperinsulinemic–euglycemic glucose clamp. Due to the complexity and low availability of the procedure (the test may only be performed in a hospital setting), it is used exclusively in scientific research [47].

The remaining trials showed no significant changes in glycemic markers. In the majority of analyzed literature (apart from one [35]) all glycemic parameters used for prediabetes diagnosis according to ADA guidelines were reported, including FBG, 2-h OGTT, and HbA1c level [31]. In some of the studies, HOMA-IR, insulin levels and *C*-peptide levels [39] were also assessed. This latter group of parameters are not listed in the ADA guidelines for diagnosing and monitoring prediabetes and diabetes, thus, for now, they are not among the standard parameters used for assessing the risk of diabetes developing. 

During the initial literature review, we noticed several methodological problems in reviewed studies, which might have affected the results reported and made it difficult to draw a conclusion. Some trials were excluded from this review due to those concerns. 

Firstly, all studies which combined participants with prediabetes and diabetes in one group were excluded. Patients with those two conditions differ significantly in their glucose metabolic profiles and should be considered as two independent study groups. Only separating patients with prediabetes from those with diabetes makes it possible to draw valid conclusions.

Secondly, the demographic characteristics of participants should be carefully considered. Vitamin D levels, glucose metabolism, and type 2 diabetes risk vary by age and ethnicity [48]. It is also worth mentioning that ethnicity and specially race is a very important factor connected with vitamin D level [49]. For the darker skin tone, due to the absorption of UV radioation by higher levels of melanin, more time of sun exposure is required in order to synthesize vitamin D [50]. This issue increases the risk of hypovitaminosis. Particular ethnic groups should be studied separately. Glucose results in groups, which are highly diverse ethnically or include patients with an extremely broad age range, e.g., 18–80 years old, may vary for reasons unrelated to the experimental treatment and thus lead to the distortion of the results. For this reason, it is important that the results are analyzed for relatively homogenous groups of patients.

The serum concentration of vitamin D is closely related to the season of the year and geographical location [51]. Only three articles we analyzed reported taking into account the effects of sun exposure or seasons [35,37,39]. This is an important factor, as people living close to the equator (who often cover themselves to protect from sun exposure) and those living far from the equator (low UVB radiation) are much more likely to be deficient in vitamin D [25]. What is more, using a sunscreen with a greater than 12 SPF (sun protection factor) prevents the production of vitamin D in the skin [25]. None of the articles included this aspect in their analysis, and it is worth taking this aspect into account when designing future randomized trials.

Many trials we reviewed did not report on the use of medications or any diseases that could potentially influence glucose or vitamin D metabolism. What is even more important, some articles did not report any exclusion criteria at all. When designing future studies, it seems important to standardize the exclusion criteria across all trials. Not only current medications, comorbid medical conditions, and the participant’s medical history are essential, but mental health disorders should also be accounted for, as mental impairment can influence compliance. These four elements should always be taken into account when formulating the exclusion criteria and should be part of the development of the trial methodology.

Another limitation that might lead to the inconsistency of the results is combining interventions in one study. Numerous randomized studies were conducted with the co-supplementation of calcium, vitamin K, or omega-3. This made it difficult to assess the independent role of vitamin D supplementation in glucose metabolism and diabetes 2 prevention [52,53]. In a few of the randomized trials we reviewed, variable doses of vitamin D or different frequencies of administration were used during one study. For instance, patients with hypovitaminosis were given large doses for a few weeks to normalize their serum level of vitamin D, and the doses were then reduced significantly for the remainder of the follow-up period [54,55]. This methodology made it impossible to determine which intervention influenced the obtained results the most. 

The last and the most important intervention, which was repeatedly combined in randomized trials with vitamin D supplementation in prediabetes populations was educating the participants about diabetes 2 and lifestyle modifications [54,56]. The recommended and most effective method for preventing diabetes 2 developing in patients with prediabetes is lifestyle behavioral change [57]. Introducing such an important additional intervention into the study design may significantly influence the results and lead to incorrect conclusions. When designing future studies, combining several interventions in one study should be avoided.

To summarize our remarks, concerns related to existing clinical study data and our recommendations for future RCT designs are listed in Table 2.

Our systematic review also has some limitations. Firstly, a meta-analysis was not completed, which would be a valuable addition to this article. This was due to the variation between the methodology in the analyzed studies. Moreover, this review was based only on articles published in the last 10 years, while similar papers had been published earlier. This was due to the need to use the most current studies based on the most up-to-date knowledge about prediabetes. In addition, the last 10 years have seen the largest number of publications on this topic. The last limitation resulted from more than two investigators reading abstracts in the 2nd stage of the review. This may have resulted in a less consistent selection of literature.

## 5. Conclusions

After reviewing the existing literature from the last 10 years, we can conclude that vitamin D supplementation in patients with prediabetes and hypovitaminosis D does not significantly influence glycemic parameters and conversion to type 2 diabetes mellitus. At this point, however, it should be emphasized that due to the great variation and biases in study designs, an unambiguous interpretation of the results is not possible. To eliminate those vulnerabilities in future studies, we make certain suggestions for study design. Although the results of the reviewed studies seem to indicate no influence of vitamin D supplementation on glucose metabolism in prediabetic patients long-term, better-designed studies are required to conclusively confirm this hypothesis. 

## Figures and Tables

**Figure 1 nutrients-15-00983-f001:**
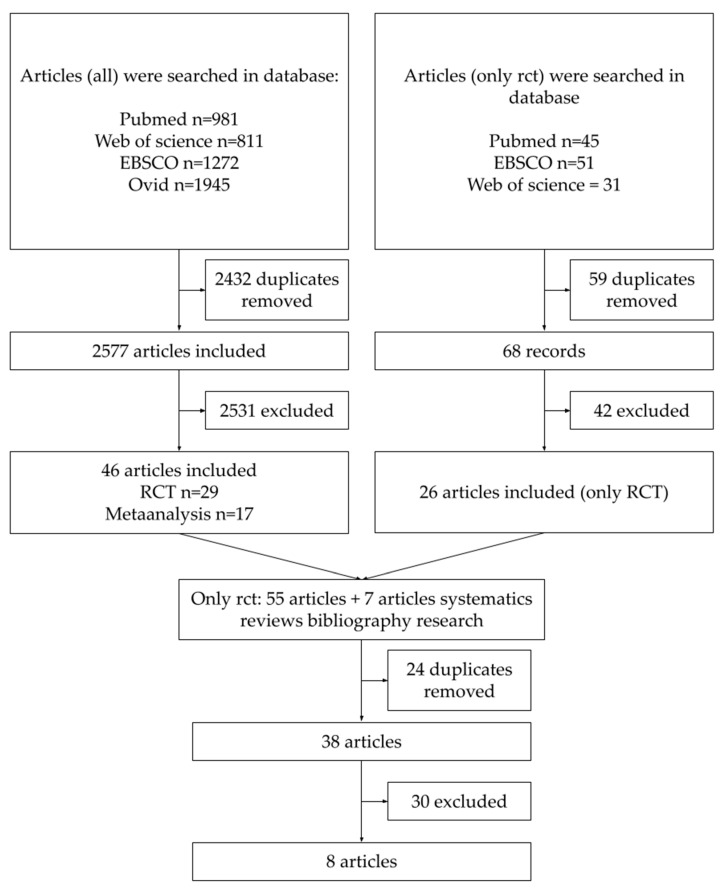
Article selection.

**Table 1 nutrients-15-00983-t001:** Interventional studies of vitamin D effects on glucose metabolism in prediabetes.

Authors	Year	No. of Participants	Time	Vit D Dose [IU]	Mean BMI	Mean 25(OH) Baseline Level	Outcome	Effect
Mohammed Al Thani et al. [33]	2019	209	6 months	4000/daily	30.0 ± 6.2	14.9 ± 4.3 ng/mL	Glucose metabolism	No change
Rolf Jorde et al. [34]	2016	511	5 years	20,000/weekly	30.1 ± 4.1	59.9 ± 21.9 nmol/L	Progression to T2DM and glucose metabolism	No change
Rasoul Zarrin et al. [35]	2017	120	3 months	1000/daily	28.71 ± 4.29	19.36 ± 13.51 ng/mL	Glucose metabolism	Improve
Anastassios G Pittas et al. [36]	2019	2423	2.5 years	4000/daily	32 ± 4.5	27.7 ± 10.2 ng/mL	Progression to T2DM and glucose metabolism	No change
Tomi-Pekka Tuomainen et al. [37]	2015	73	5 months	1600 or 3200/daily	29.4 ± 2.7	57.0 ± 11.0 nmol/L	Glucose metabolism	No change
Mayer B. Davidson et al. [38]	2013	117	12 months	~88,865/weekly	32.9 ± 4.3	22.0 ± 4.8 ng/mL	Glucose metabolism	No change
Helen J Wallace et al. [39]	2019	66	26 weeks	3000/daily	34.7 ± 8.0	30.7 ± 14.3 nmol/L	Glucose metabolism	No change
Tracy S. Moreira-Lucas et al. [40]	2016	72	24 weeks	28,000/weekly	30.1 ± 3.9	48.1 ± 14.3 nmol/L	Glucose metabolism	No change

**Table 2 nutrients-15-00983-t002:** Current concerns and recommendations for future RCT’s designing.

No.	Concerns Related to Existing Clinical Study Data	Recommendations for Future RCT’s Designing
1.	Follow-up too short	Follow-up > 12 months
2.	Too high, too low, non-physiological, and not frequent enough doses of vitamin D	Sufficiently large doses, consistent with the recommendations for a given population
3.	Lack of reported exclusion criteria or a wide variety of exclusion criteria inconsistent between studies	Medications, current medical diseases, participant’s past medical history, which could potentially influence glucose or vitamin D metabolism, and mental illnesses routinely included in the exclusion criteria beside any relevant others
4.	Different endpoints and glycemic parameters measured	Ideal primary endpoint: diabetes mellitus type 2 developmentThe measured parameters should be consistent with prediabetes criteria established by scientific societies
5.	Vitamin D levels in participants not assessed or participants with normal levels of vitamin D included in study groups together with vitamin D deficient participants	Only participants with hypovitaminosis D divided into levels of this deficiency should be considered in order to get reliable results
6.	Study groups not homogenous regarding: diabetes status	Only patients with diabetes or with prediabetes in one study group
7.	Wide age-range and different ethnicity	Homogenous groups analyzed together
8.	Additional interventions in one study	Vitamin D supplementation should be the only intervention; neither lifestyle change nor co-supplementation should be included in study designs, not allowing for separating the effects of individual interventions
9.	Not reporting or assessing sun exposure, the seasons, and geographical location in which the study was performed	Inclusion of sun exposure, seasons, and geographical locations data with respect to vitamin D and participants

## Data Availability

All data are avaible upon request.

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
