# Peer review of "Controversial Impact of Vitamin D Supplementation on Reducing Insulin Resistance and Prevention of Type 2 Diabetes in Patients with Prediabetes: A Systematic Review"

_nutrients, 2023, doi:10.3390/nu15040983_

Round 1
Reviewer 1 Report
Thank you for the opportunity to read this interesting review. Overall, I think there are valuable insights to other researchers in the field from this research study.
I do think there are issues that need to be addressed prior to publication.
Abbreviations/acronyms should be defined prior to first usage - WHO, ADA etc
Abstract: check grammar and a bit more detail in methods please.
Scientific manuscripts should not write "most studies" - be specific - what percentage?
Also, in the introduction and abstract the authors state they want to explore why study results were varied - but in the discussion they state the purpose of the review was to evaluate the current knowledge - these are two different research questions, though they are related, please clarify.
Lines 55-56 - needs evidence/reference to support
Line 58 - catalyst?
Line 62 - economies? consider different word choice
Line 65 - definitions of deficiency vary by country - and 50nmol/L is most definitely not universally agreed - and may actually be part of the reason for study discrepencies - please clarify and discuss this point
Line 66 - vitamin D deficiency in adults causes osteomalacia not osteoporosis - to be honest a rather disturbing mistake to read in a paper reviewing vitamin D
lines 75-77 - vague - please clarify
line 70 - needs reference
Methods- please explain why 30 studies were excluded and why the 8 were included
Also, was peer review one of the inclusion criteria? please state in abstract and early in methods
line 111- what does "conflicts" mean?
Lime 184 - rather than "most" please be specific
line 185 - what does "all of them" refer to?
when stating "6 studies" please specify studies
line 218 - rather than "most" please specify percentage
A major issue I have with this manuscript is the following: many suggestions are useful for future studies, but there is one issue the authors have not considered - when they state that studies should last an adequate amount of time AND include patients with vitamin D hypovitaminosis they are not considering the ethical issues of not treating people who have a deficiency (as the most robust studies will have a control group) - can the authors suggest a way to conduct a study with a control group that is not unethical?
Lastly, the table shown in the conclusion belongs in the discussion
Author Response
Dear Sir,
on behalf of myself and the other authors, I would like to thank you for your comments. Firstly, I’d like to reffer to the ethical concerns raised in your comment. All studies included in our systematic reviews had ethics committee approvals, although most of them included patients with vitamin D deficiency in the control groups. Vitamin D supplementation in the form of tablets/capsules is not the main source of its supply. All study participants have access to sun exposure and food, which are the main sources of vitamin D. However, if there were still ethical doubts, two solutions could be applied:
- In the control group and the research group, vitamin D supplements should be used, but with different doses(larger/smaller). Which, unfortunately, will certainly affect the conclusions of such a study.
- Participants with a hipovitaminosis D of between 30 nmol/L-50 nmol/L should be included in the study. According to the National Institutes of Health(NIH), only level below 30 nmol/L can affect health.
The second point I would like to refer is the issue of peer review. We did our systematic review on the basis of four main scientific data bases i.e. Pubmed, EBSCO, Web of science and Ovid. They publish articles from journals with a high impact factor, with peer review as an obligatory aspect. For this reason, we did not consider it necessary to include them as an inclusion criterion. We addressed all other suggestions in the article and corrected them. Thank you very much for your interest in our work and for such detailed recommendations for its improvement. It certainly had a positive impact on the quality of our article.
Please see the attachment for details.
Yours faithfully
Agata Pieńkowska

Reviewer 2 Report
The authors have examined available studies to determine the potential benefits of vitamin D supplementation in preventing the conversion of prediabetes to diabetes. The manuscript has been well structured and written. Following are a few minor comments:
In Fig. 1, exclusion criteria at various levels may be given in the legend.
Font size of Fig 1 needs to be increased. Boxes may be removed so that the font size is not limited.
Results: Outcome (Progression to T2DM and glucose metabolism) may be given in detail for the 8 studies that were studied/tabulated in Table 1.
Author Response
Dear Sir,
on behalf of myself and the other authors, I would like to thank you for your comments and such positive reviews. We did not include detailed results in table 1 due to the fact that each study was based on different parameters of glucose metabolism, which made the table difficult to read. In addition, for this reason, we do not discuss the results in such detail and do not perform meta-analysis. We have made corrections to scheme 1 as far as possible to keep it legible.
Please see the attachment for details.
Yours faithfully
Agata Pieńkowska
